# OpenReview forum: "Improving Factuality and Reasoning in Language Models through Multiagent Debate"
_ICLR.cc/2024/Conference — Submitted to ICLR 2024_

### Official Review · Reviewer_Kfe5 · 2023-10-28

**Soundness:** 2 fair
**Presentation:** 3 good
**Contribution:** 2 fair
**Rating:** 5
**Confidence:** 4

**Summary:**

This paper introduces a new approach to improve the language responses of Large Language Models (LLMs) by implementing a multiagent debate system. The authors propose a method where multiple language model instances propose and debate their individual responses over multiple rounds to arrive at a unified final answer. The paper demonstrates that this approach significantly enhances mathematical and strategic reasoning and improves the factual validity of generated content. The authors also introduce a new benchmark for evaluating the factual accuracy of generated biographies of famous computer scientists.

**Strengths:**

1. This paper presents a new approach to improve the performance of Large Language Models (LLMs) using a multiagent debate system.
2. The introduction of a new benchmark for evaluating the factual accuracy of computer scientist biographies is an addition to the field.

**Weaknesses:**

1. The novelty is overstated. In essence, the primary experiment merely involves using different random seeds, prompting ChatGPT to generate varied responses, and then refining them. Although there's an additional experiment that uses different language models like ChatGPT and Bard, the test set is very limited.

2. The proposed method is resource-intensive and not suitable for lengthy questions or answers. Employing multiple agents and several rounds of debate can lead to very long conversations, potentially exceeding the context limit and reducing performance.

3. While the presentation is drawn out, the idea and approach are straightforward. Nevertheless, the author dedicates considerable space to prompt examples, and the explanations are protracted.

**Questions:**

None

---

> ### Author Response · Authors · 2023-11-18
> **Rebuttal Response**
>
> Thank you for your detailed comments. Please see our clarifications below.
>
> **Novelty.** The primary novelty of the paper is proposing the methodology of multi-agent debate as an inference-time approach to improving the performance of LLMs across both factuality and reasoning. While prior works have explored how both using ensembles of models improves performance as well using self-reflection of a single model, our paper illustrates how both ideas may be jointly combined to generate more accurate responses.
>
> This combination goes beyond simply having an ensemble of self-reflection agents, as the debate procedure lets each agent reflect on both the generations of the original language model and that of other models. We illustrate the power of the debate procedure in new Figure XXXI ([screenshot here](https://ibb.co/426vcgb)) and Figure XXXII ([screenshot here](https://ibb.co/0FSWZFz)) in the updated paper, where we show how debate can be used to self-correct two initially incorrect answers.
>
> We have further provided an additional experiment using debate on the open-source chat-Llama 7B model. Below, we find that similarly, debate improves final performance.
>
>  | Model | Arithmetic | GSM  | MMLU |
> | -------- | ------- | ------- | ------- |
> |Single Agent  |  9.0 +- 1.6  | 20.7 +- 2.3 | 41.0 +- 2.8 |
> |Single Agent (Reflection) | 10.7 +- 1.7 | 21.0 +- 2.3  | 39.7 +- 2.8 |
> |Multi-Agent (Majority) | 11.0 +- 1.8 |  25.7 +- 2.5 | 43.3 +- 2.9|
> |Multi-Agent (Debate) | 13.3 +- 1.9  | 29.3 +- 2.6 | 47.7 +- 2.9  |
>
> We have added this to Appendix A.1 of the paper.
>
> **Computational Expense.**  While our approach is computationally expensive compared to a single forward pass to a language model, it can be used as a method to take a computationally cheap model (i.e. GPT-3.5) and have it perform similarly to a significantly more computationally expensive model (i.e GPT-4 which is over 100 times more expensive), providing computational savings. In addition, the multiagent debate procedure can be distilled down into another model both for faster inference as well as a method for self-improvement in a language model.

---

> > ### Author Response · Authors · 2023-11-21
> > **Author Followup**
> >
> > Dear Reviewer Kfe5,
> >
> > Thank you for your time spent reviewing our paper. We have tried to clarify your concerns on novelty and clarity. As the discussion period ends tomorrow, please let us know if you would like additional clarifications.
> >
> > Thanks,
> >
> > Paper Authors

---

### Official Review · Reviewer_CMZe · 2023-10-31

**Soundness:** 3 good
**Presentation:** 3 good
**Contribution:** 3 good
**Rating:** 5
**Confidence:** 5

**Summary:**

This paper proposes to improve factuality and reasoning in LLMs by employing multiple agents to independently generate reasoning paths, and then leverage responses from each other to derive the final response across multiple rounds. Evaluated on math and reasoning benchmarks, as well as factuality benchmarks, results suggest that the proposed method can outperform previous methods including self-consistency.

**Strengths:**

1. This paper introduces an interesting and effective method utilizing multiple agents which reference each other to derive answers for the next round. Compared to similar methods (self-consistency), the proposed method achieves better performance on various benchmarks. This can be an interesting finding to future research in prompting and leveraging generation from multiple LLMs for self-critic.
2. The paper presents several insightful ablation studies, suggesting that different language models, and different prompts can complement with each other. This finding may suggest a new direction in inspecting knowledge obtained in LLMs and uncertainty represented.

**Weaknesses:**

1. Although the performance improved, it is still not clear why "debating" would improve model performance. Especially, for uncertain answers (examples in Figure 9), it is not convincing why referencing each others' answers would generate more factual answers, as reasoning is not very relevant. Further exploration is required, in particular with more agents. Moreover, even for reasoning tasks (in Figure 2 Round 2), it does not seem that much reasoning is involved. The model is mostly copying, or echoing what the correct answer is, rather than having in-depth "debate", or reasoning, about why one answer is correct and the other is wrong. Without more analysis, this contribution is limited.
2. Following the first point, some details are not clear. See questions below.

**Questions:**

1. Can you clarify what short and long represent in Figure 3?
2. Can you explain why "summarization" would actually improve the model performance? Inevitable summarization would result in information loss. This finding seems counter-intuitive. Have you conducted similar experiments on other tasks?
3. When would the model agents be more "agreeable"? Would they be more agreeable when there are more similar answers (regardless of correctness), or would then tend to be more agreeable when the answer is correct? Can you verify the relationship between "agreeable" and "a result of instruction tuning or RLHF" using pre-aligned checkpoints?

---

> ### Author Response · Authors · 2023-11-18
> **Rebuttal Response**
>
> Thank you for your valuable feedback on the manuscript. We have responded to each concern you’ve mentioned below.
>
> **Short vs Long.** In Figure 3, short refers to a prompt to a large language model that induces debates to be shorter in length, while long refers to a prompt to a large language model that induces debates to be longer in length. We have updated the paper to make this clearer.
>
> **Understanding Gains in Debate.**  The Multiagent Debate procedure in the paper tightly couples two sources of performance gains in LLMs, (1) wisdom of the crowds, where multiple instances of LLMs can jointly leverage to improve performance (i.e. from majority voting)  (2) self-reflection, the ability for LLMs to use previous generations to improve their performance. However, the procedure goes beyond directly combining these two aspects because individual LLMs can reflect not only based on their thought process but also that of other agents (which is referred to as debate in the paper). The core reason multiagent debate improves This allows LLMs to recover and generate correct answers in a subsequent round of reasoning even if all answers in the current round are incorrect. This is illustrated in Figure 11 ([screenshot here](https://ibb.co/YDM7ZGc)) of the original paper and in Figure A.XXXI ([screenshot here](https://ibb.co/426vcgb)) and A.XXXII ([screenshot here](https://ibb.co/0FSWZFz)) in the updated paper.
>
> To more clearly understand the performance of debate, we also consider a new baseline of running ensembles over self-reflection, where we instantiate multiple instances of a self-reflection agent and then take a final ensemble vote over generated self-reflection results.  We find that debate outperforms this baseline, indicating that discourse between agents is helpful overall beyond ensembling answers from a self-reasoning language model.
>
>  | Model | Arithmetic | GSM  | MMLU |
> | -------- | ------- | ------- | ------- |
> |Single Agent  | 67.0 +- 4.7  | 77.0 +- 4.2 | 63.9 +- 4.8 |
> |Single Agent (Reflection) | 72.1 +- 2.4  | 75.0 +- 4.3 | 57.7 +- 5.9 |
> |Multi-Agent (Majority) | 75.0 +- 3.9 | 81.0 +- 3.9  | 67.0 +- 4.7 |
> |Multi-Agent (Reflection) | 76.0 +- 4.3 | 80.0 +- 4.0 | 65.0 +- 4.7 |
> |Multi-Agent (Debate) | 81.8 +- 2.3 | 84.0 +- 2.1  | 71.1 +- 4.6 |
>
> To further understand the performance of debate with more agents, we also consider using a total of 10 agents for debate across 2 rounds of debate in the table below compared to majority vote over 50 agents. We find that having more agents in debate continues to improve the performance.
>
>
> | Model | Arithmetic | GSM  | MMLU |
> | -------- | ------- | ------- | ------- |
> | Majority Vote (50 agents) | 92.0+- 2.7  | 85.0 +- 3.6 |  67.0 +- 4.7 |
> | Debate (10 agents 2 rounds) | 96.0+- 1.9 |  89.0 +- 3.1 | 71.0 +- 4.5 |
>
> **Gains through Summarization.** We found that when context lengths became long, LLMs would struggle to be able to process the entire length of the message (also discussed in [1]).   Summarization serves as a way to remove extraneous repeated information generated by each agent to condense relevant information to solve the task. In addition to results on the arithmetic task, we ran additional experiments and found that summarization on GSM leads to a performance of 87.0 compared to the original performance of 85.0 with 3 agents and 2 rounds of debate. We found that summarization on MMLU leads to a performance boost of  73.0 compared to the original performance of 71.1.  We have added these experiments to Appendix A.1
>
> **Analysis of Agreeableness.** In our experiments, we found that agents were agreeable regardless of whether they knew the answer was correct so long as there were similar answers from other agents. To verify the relation between “agreeable” and “a result of instruction tuning or RLHF”, we consider the LLama-2 7B and the RLHF chat-LLama-2 7B models. On the GSM8K task, we find that  LLama-2 7B model achieves a consensus of 51.3% after debate compared to consensus of 62.7% with aligned chat-LLama 2 model. On the MMLU task, we find that  LLama-2 7B model achieves a consensus of 74.3% after debate compared to consensus of 100.0% with aligned  chat-LLama 2 model. This supports the conclusion that RLHF models tend to be more agreeable than the previous unaligned checkpoints. We have added this experiment to Appendix A.1.
>
> [1] Liu et al. Lost in the Middle: How Language Models Use Long Contexts

---

> > ### Comment · Reviewer_CMZe · 2023-11-22
> >
> > Thanks for the responses.
> >
> > Can you clarify what reasoning procedure is done from my comments above? I'm still not convinced by your response on what exactly the models are leveraging from each other to improve their reasoning capabilities before deriving the final answer.

---

> ### Author Response · Authors · 2023-11-21
> **Author Followup**
>
> Dear Reviewer CMZe,
>
> Thank you for your time reviewing the paper. We have run additional evaluations to address the concerns you raised earlier in your response. As the discussion period draws to a close tomorrow, please let us know if there is a need for additional clarification.
>
> Thanks,
>
> Paper Authors

---

> ### Author Response · Authors · 2023-11-22
> **Response**
>
> Thank you for your response.
>
> The reasoning procedure that occurs during multiagent debate is the use of different lines of reasoning from separate agents to stitch together an overall response. When generating initial answers, different agents provide different approaches to solve a problem, where each approach has errors in different places in its reasoning. Self-refinement through multiagent debate allows each agent to see both their original solution and the solution of other agents and allows an agent to stitch together the correct parts of reasoning in its original solution with the correct parts of reasoning in other agents' solutions. This allows an agent to turn two initially incorrect answers to a fully correct one.
>
> We illustrate one example of this process in Figure 11 ([screenshot here](https://ibb.co/YDM7ZGc)) of the paper. In this example, in its initial response, the chatGPT agent has a flaw in its reasoning where it does not realize that Carla needs to fully redownload a file.  At the same time, in its initial response, the Bard agent realizes that Carla needs to fully redownload a file, but incorrectly computes the time it takes to download a file. Through the multiagent debate procedure, the chatGPT agent can take the correct path of reasoning from Bard in combination with its own reasoning to synthesize a correct solution to the full problem.
>
> Another example of this can be found in the newly added Figure XXXI ([screenshot here](https://ibb.co/426vcgb)) and Figure XXXII ([screenshot here](https://ibb.co/0FSWZFz)). In Figure XXXI ([screenshot here](https://ibb.co/426vcgb)), in the first round, both agents make different mistakes in evaluating the arithmetic expression, where the first agent incorrectly adds each component together at the end and the second agent carries over the wrong answer from a previous computation. Both agents stitch the reasoning of the other agent with their original reasoning in the previous round to get the final answer correctly.

---

### Official Review · Reviewer_JxCs · 2023-11-01

**Soundness:** 3 good
**Presentation:** 2 fair
**Contribution:** 3 good
**Rating:** 6
**Confidence:** 4

**Summary:**

This paper proposes a method called multi-agent debate to improve the performance of LLMs on factuality and reasoning.

The paper contributes:
1. A novel method that is multi-agent debate;
2. Claimed by the authors, a new benchmark of factual correctness that language models struggle with;
3. Extensive experiments are conducted to verify the effectiveness of the proposed method, based on six different reasoning and factual accuracy tasks.

**Strengths:**

Originality: The proposed method is novel. Though some similar and more advanced ideas have come up recently, this paper should be the very first paper that studies multi-agent ensemble + reflection.

Significance: The experiments of the paper are extensive. The method has been tested on several reasoning tasks.

**Weaknesses:**

Evaluation: The paper lacks a deep evaluation of why multi-agent debating can improve the LLMs' reasoning ability.

Based on my understanding, multi-agent debating = ensemble multiple model's answers + models' evaluation.

However, the paper does not analyze the intrinsic reason why the method can work. Although the analysis in the paper explores the impact of different numbers of agents and rounds of debate on the results, this outcome is predictable, as having more debates and rounds would yield more answers. By ensembling these results, the performance would naturally improve. However, such analysis still does not touch upon the essence of the method. It may be an issue with how the paper is written, as it keeps emphasizing the term 'debate', obscuring the fundamental principles of the method.

Lacking scaling-up experiments: One baseline called majority voting is important since it also ensembles multiple model answers. One issue of the baseline explored in this paper is that it only uses around 3*2 =6 answers, which is quite small. It is interesting to see how effective the multi-agent debate is when compared to majority voting with 50 or even 100 samplings.

**Questions:**

Why the paper didn't use different model bases to study multi-agent debate? Mostly, the paper only uses one gpt-3.5 or gpt-4 as the base. I know PaLM is used in the further evaluation experimental part, but it is not enough.

---

> ### Author Response · Authors · 2023-11-18
> **Rebuttal Response**
>
> Thank you for your valuable feedback on the manuscript. We respond to each concern you have listed above below and have also updated the paper.
>
> **Analysis of Performance Gains From Debate.** The Multiagent Debate procedure in the paper tightly couples two sources of performance gains in LLMs, (1) wisdom of the crowds, where multiple instances of LLMs can jointly leverage to improve performance (i.e. from majority voting)  (2) self-reflection, the ability for LLMs to use previous generations to improve their performance. However, the procedure goes beyond directly combining these two aspects because individual LLMs can reflect not only based on their own thought process but also that of other agents (which is referred to as debate in the paper). The core reason why this helps performance is that now LLMs can stitch together correct paths of reasoning in one response with that of other responses.   This allows the debate procedure to take a round of responses where all answers are incorrect, and recover answers in the subsequent round that are fully correct, by fixing misunderstandings/errors in reasoning using responses from other agents. This is illustrated in Figure 11 ([screenshot here](https://ibb.co/YDM7ZGc)) of the original paper and in Figure A.XXXI ([screenshot here](https://ibb.co/426vcgb)) and A.XXXII ([screenshot here](https://ibb.co/0FSWZFz)) in the updated paper.
>
> To more clearly differentiate the performance of debate, and simply running ensembles over self-reflection, we consider a new baseline, where we instantiate multiple instances of a self-reflection agent and then take a final ensemble vote over generated self-reflection results.  We find that debate outperforms this baseline, indicating that discourse between agents is helpful overall simply just ensembling answers from a self-reasoning language model.
>
>  | Model | Arithmetic | GSM  | MMLU |
> | -------- | ------- | ------- | ------- |
> |Single Agent  | 67.0 +- 4.7  | 77.0 +- 4.2 | 63.9 +- 4.8 |
> |Single Agent (Reflection) | 72.1 +- 2.4  | 75.0 +- 4.3 | 57.7 +- 5.9 |
> |Multi-Agent (Majority) | 75.0 +- 3.9 | 81.0 +- 3.9  | 67.0 +- 4.7 |
> |Multi-Agent (Reflection) | 76.0 +- 4.3 | 80.0 +- 4.0 | 65.0 +- 4.7 |
> |Multi-Agent (Debate) | 81.8 +- 2.3 | 84.0 +- 2.1  | 71.1 +- 4.6 |
>
>
> **Scaling Up Agents.** We compare running multiagent debate over 10 agents over 2 rounds with majority vote across 50 agent responses. We find that even with a very large number of agents, running multiagent debate still outperforms using a majority vote.
>
> | Model | Arithmetic | GSM  | MMLU |
> | -------- | ------- | ------- | ------- |
> | Majority Vote (50 agents) | 92.0+- 2.7  | 85.0 +- 3.6 |  67.0 +- 4.7 |
> | Debate (10 agents 2 rounds) | 96.0+- 1.9 |  89.0 +- 3.1 | 71.0 +- 4.5 |
>
> **Other Model Bases.** We have added additional results using the chat-Llama 2 7B model. Below we report the performance of multiagent debate and each baseline. We find that multi-agent debate is also helpful on open-source language models.
>
>  | Model | Arithmetic | GSM  | MMLU |
> | -------- | ------- | ------- | ------- |
> |Single Agent  |  9.0 +- 1.6  | 20.7 +- 2.3 | 41.0 +- 2.8 |
> |Single Agent (Reflection) | 10.7 +- 1.7 | 21.0 +- 2.3  | 39.7 +- 2.8 |
> |Multi-Agent (Majority) | 11.0 +- 1.8 |  25.7 +- 2.5 | 43.3 +- 2.9|
> |Multi-Agent (Debate) | 13.3 +- 1.9  | 29.3 +- 2.6 | 47.7 +- 2.9  |
>
> We have also added this table in Appendix A.1

---

> > ### Author Response · Authors · 2023-11-21
> > **Author Followup**
> >
> > Dear Reviewer JxCs,
> >
> > Thanks so much for your time reviewing the paper. As the discussion period is drawing to a close tomorrow, we wanted to check if our additional experiments address your concerns. If not, we are happy to add additional evaluations or clarifications.
> >
> > Thanks,
> >
> > Paper Authors

---

> > > ### Comment · Reviewer_JxCs · 2023-11-23
> > >
> > > Thanks for the additional experimental results.
> > >
> > > But I would keep my original score since the paper writing on the explanation of why the method works is still not good enough.

---

### Official Review · Reviewer_ML3o · 2023-11-05

**Soundness:** 4 excellent
**Presentation:** 4 excellent
**Contribution:** 3 good
**Rating:** 8
**Confidence:** 3

**Summary:**

The paper introduces a method where multiple language model instances individually propose and jointly debate their
responses and reasoning processes to arrive at a common answer - essentially a form of voting applied to language models. They test this on six datasets, some of which are datasets that contain challenging reasoning questions.

Interestingly, the authors note that the various answers, following a debate, converge on a single common answer.

**Strengths:**

A novel approach is presented that raises the performance of LLMs through inter-model debate. This approach is subject to a number of parameters (#agents, #rounds of debate etc.), and the effects of these parameters (e.g., debate length on accuracy) are carefully analyzed, and it is reported how other approaches, such as chain-of-thoughts, improve this approach. The write-up is very clear, and the appendix contains a host of useful pieces of information, such as examples, performance scores, dataset descriptions etc.

**Weaknesses:**

(See "Questions" section)

**Questions:**

In the paper where the Minerva model was introduced (https://arxiv.org/abs/2206.14858), which you also cite in terms of using fine-tuning, the authors are using a simple form of majority voting (on a single model) to enhance the output performance.
Do you know how much your multi-model approach improves over this form of "auto-debate" (i.e., if you run for each of the models' majority voting, like in the Minerva paper, how close would that come to your results)?

You have mentioned further majority voting schemes in the Related Work section. Could you perhaps insert a table highlighting the differences and similarities of your approach to these approaches? You essentially say: "In contrast, in our work, we aim to use communication between different language models to enable more effective reasoning and factuality in language models." Since your methodology has a number of parameters (#agents, #rounds of debate etc.), it would be good to have a more comprehensive comparison.

If these questions are answered in full, I'd be happy to consider raising my rating.

---

> ### Author Response · Authors · 2023-11-18
> **Rebuttal Response**
>
> Thank you for your valuable feedback on the manuscript. We have addressed each issue you have listed below and have also updated the paper.
>
> **Majority Voting in Minerva.** We would like to clarify that the multiagent majority baseline we include in Table 1 and 2 of the paper  also corresponds to the multiagent majority (auto-debate) baseline in Minerva. We generate multiple answers using a single instance of a language model and use the majority vote to determine the final answer. We have updated the description of the multiagent majority baseline to clarify this in Section 3.1  and have also added an additional reference to the Minerva paper there and in the related work.
>
> **Multiagent Majority in Related Work.** We have added Table 3 in the related work comparing the difference between the various multiagent majority voting baselines and debate. The primary difference is that we use multiple rounds of communication between agents to achieve the final answer while multiagent majority simply takes the majority vote at the end of the first round (no discourse between models). Both approaches have a flexible number of agents $N$, while the rounds of debate is 1 in multiagent majority and $T$ in multiagent debate.

---

> > ### Author Response · Authors · 2023-11-21
> > **Author Followup**
> >
> > Dear Reviewer ML3o,
> >
> > Thanks so much for your time reviewing the paper. As the discussion period is drawing to a close tomorrow, we wanted to check if our response answers your question in full. If not, we are happy to add additional clarifications.
> >
> > Thanks,
> > Paper Authors

---

> > > ### Comment · Reviewer_ML3o · 2023-11-23
> > >
> > > I thank the authors for their effort in revising the paper, and I see a number of items were added as clarifications. I am happy that this constitutes important work on improving LLMs, so I am updating my score to "accept".

---

### Author Response · Authors · 2023-11-21
**General Response**

We thank the reviewers for their thorough feedback. Reviewers noted that the write-up is clear (Reviewer ML3o) and interesting and new (Reviewer ML3o, JxCs, CMZe, Kfe5). Reviewers had some questions about the source of gains in multiagent debate and requested additional experimental evaluations. To address this, we have updated the paper in the following ways:

- To clarify the effect of multiagent debate, we compare our approach with ensembling a set of self-refining agents in Table VII and illustrate how it outperforms this approach. We have further added two qualitative illustrations of debate when the original answers of both agents are incorrect in [Figure XXXI](https://ibb.co/gr69Z4m) and [Figure XXXII](https://ibb.co/FxsGK75), illustrating the changes in responses induced by debate.
- To clarify how our approach differs from various multiagent majority baselines, we’ve added Table 3 in the related work and a discussion in the method section.
- We’ve added an additional experiment in Table V illustrating how our approach outperforms majority voting with 50 agents.
- We’ve added an additional experiment illustrating how multiagent debate can be applied to the chat-Llama 2 7B model and also improve performance
- We’ve added an experiment illustrating the agreeableness of language models after RLHF finetuning in Appendix Section A.1.
- We've illustrated how summarization helps performance on other tasks in Appendix Section A.1

Changes in the paper are highlighted in blue. For additional changes and per reviewer feedback, please also see our individual response. Please let us know if there are any additional questions or concerns about the paper.

---

### Meta-Review · Area_Chair_zeRr · 2023-12-06

**Metareview:**

The paper introduces a novel method of 'multiagent debate' to improve the performance of language models. This approach enables different agents to stitch together the most logical parts of their reasoning, leading to more accurate solutions. Key strengths include the originality of the concept and extensive experiments demonstrating its effectiveness. Weaknesses lie in the lack of deep evaluation of why this method outperforms others, and the method's resource-intensive nature which may not be practical for longer questions or responses.

**Justification For Why Not Higher Score:**

The decision to not award a higher score hinges on the unresolved concerns about the fundamental reasoning process behind the multiagent debate, as highlighted by Reviewer CMZe. Additionally, the method's practicality and computational intensity, as pointed out by Reviewer Kfe5, limit its broader applicability, which is crucial for a higher evaluation.

**Justification For Why Not Lower Score:**

N/A

---

### Decision · Program_Chairs · 2024-01-16

Reject